# Three-Dimensional Investigations of Virus-Associated Structures in the Nuclei with White Spot Syndrome Virus (WSSV) Infection in Red Swamp Crayfish (*Procambarus clarkii*)

**DOI:** 10.3390/ani12131730

**Published:** 2022-07-04

**Authors:** Yovita Permata Budi, Li-Chi Lin, Chang-Hsien Chung, Li-Li Chen, Yi-Fan Jiang

**Affiliations:** 1Graduate Institute of Molecular and Comparative Pathobiology, School of Veterinary Medicine, National Taiwan University, Taipei 10617, Taiwan; f08644009@ntu.edu.tw; 2School of Veterinary Medicine, National Taiwan University, Taipei 10617, Taiwan; victoriaandjasmine@gmail.com (L.-C.L.); b07609076@ntu.edu.tw (C.-H.C.); 3Institute of Marine Biology, National Taiwan Ocean University, Keelung City 20224, Taiwan; joechen@ntou.edu.tw; 4Center of Excellence for the Oceans, National Taiwan Ocean University, Keelung City 20224, Taiwan

**Keywords:** virus assembly, ultrastructure, electron tomography, serial section, morphogenesis

## Abstract

**Simple Summary:**

Infections of white spot syndrome virus could be a lethal disease in shrimp culture systems. The 3D structures of the virus-associated structures were obtained through electron tomography. The relationships between the mature and immature particles were connected by the localizations of viral proteins. Our study provided new structural information on the virus through 3D, which extends the limited 2D knowledge of pathogen assembling in the white spot disease, further contributing to the development of a strategy for therapy or prevention.

**Abstract:**

White spot syndrome virus (WSSV) has been reported to cause severe economic loss in the shrimp industry. With WSSV being a large virus still under investigation, the 3D structure of its assembly remains unclear. The current study was planned to clarify the 3D structures of WSSV infections in the cell nucleus of red swamp crayfish (*Procambarus clarkii*). The samples from various tissues were prepared on the seventh day post-infection. The serial sections of the intestinal tissue were obtained for electron tomography after the ultrastructural screening. After 3D reconstruction, the WSSV-associated structures were further visualized, and the expressions of viral proteins were confirmed with immuno-gold labeling. While the pairs of sheet-like structures with unknown functions were observed in the nucleus, the immature virions could be recognized by the core units of nucleocapsids on a piece of the envelope. The maturation of the particle could include the elongation of core units and the filling of empty nucleocapsids with electron-dense materials. Our observations may bring to light a possible order of WSSV maturation in the cell nucleus of the crayfish, while more investigations remain necessary to visualize the detailed viral–host interactions.

## 1. Introduction

Since its outbreak was reported in Asia in the early 1990s, white spot disease (WSD) has caused huge economic losses in the crustacean farming industry throughout the world [1,2]. The disease has spread among shrimps, crabs, crayfish, and lobsters, with the highest cumulative mortality of the disease reported in cultured shrimps (up to 100% within 2–10 days of infection) [3]. The pathogen, white spot syndrome virus (WSSV), is an enveloped and rod-shaped virus containing a genome of double-strain DNA with a length of about 300 kbp [4]. In the genome, there are about 181 proteins predicted to be expressed, including more than 58 structural proteins [5,6]. 

It has been revealed in the literature that the life cycle of WSSV includes the entry of viruses through endocytosis, escaping from endosomes, viral replication in the nucleus, the inhibition of host apoptosis, and finally the disruptions of the host cells [5,7]. The matured virions were enveloped particles in bacilliform shapes, with approximately 250–380 nm in length and 70–150 nm in width, containing a thread-like extension at one end of the virion [8,9,10,11,12]. Since the investigation of virus assembling could aid the development of therapeutic strategies, viral replication and particle assembling have been widely studied at the molecular level [5]. However, due to the complexities of the virus, the connections between viral proteins and 3D cellular structures for virus assembling and morphogenesis are still unclear. 

The development of the techniques of electron microscopy has played a major role in the study of the structures of viruses [13]. Previous studies have also revealed 2D features of WSSV assembly at the ultrastructural level, yet the exact steps for virus assembling and their relationships with cellular structural changes remain to be clarified [12,14,15]. While conventional ultrastructural investigations on WSSV have provided better throughput for 2D structural screening, the 3D reconstructions of specific structures could further enhance the accumulation of structural knowledge by acquiring the missing dimensions [13,16]. The present study was thus designed to investigate 3D arrangements of the virus-associated structures in the WSSV-infected nucleus. After the establishment of the 3D models, the virus protein expressions on virus-associated structures were further explored with immuno-gold labeling in this study.

## 2. Materials and Methods

### 2.1. Animals

Red swamp crayfish (*Procambarus clarkii*) captured in the area near Taipei city were continuously maintained in water tanks for more than 2 generations in the lab under a neutral light cycle. Adult (about 20 g), intermolt male crayfish with normal behavior were selected for inoculations. The selected crayfish were isolated in water tanks of 30 cm (L) × 20 cm (W) × 25 cm (H). Food was supplied daily and 1/3 of water was changed weekly. The crayfish were acclimated in the tank for 7days before inoculation.

### 2.2. Virus Preparation and Negative Stain

The original WSSV strain was collected from WSSV-infected *Penaeus monodon* in Taiwan [11]. The virus was then proliferated and purified in *Procambarus clarkii* as described in previous studies [6]. To visualize the virus particles, negative staining was performed on the 150-mesh grids with carbon films. Glow discharge was applied to the film to make a hydrophilic surface. Then, 10 μL of the virus solution was dropped onto the carbon films for 1 min absorption. After blotting out the excess solution, the grids were quickly rinsed and blotted through 3 drops of distilled water and stained with 0.35% uranyl formate for 1 min. The grids were allowed to be air-dried after the blotting of staining solutions. Images were taken under the transmission electron microscopy (TEM) operating at 120 kV (FEI Tecnai G2 TF20 Super TWIN).

### 2.3. Virus Inoculation and Perfusion Fixation

The virus solution was diluted in PBS (0.1 mL; 1:100 dilution) and injected intramuscularly into healthy crayfish (about 20 g) between the second and third abdominal segments [6]. On day 7 after injection, the crayfish with signs of infection were subjected to TEM sample preparations. A fixative for morphological observation contained 4% (*w*/*v*) paraformaldehyde (PFA) and 2.5% (*w*/*v*) glutaraldehyde (GA) in 0.1 M sodium phosphate buffer (PB, pH 7.3), while for immune-gold labeling, a fixative containing 4% (*w*/*v*) PFA and 0.25% (*w*/*v*) GA in 0.1 M PB (pH 7.3) was used. 

The procedures for perfusion were modified from the literature [17]. Before perfusion, the crayfish were stunned in ice-cold water. The fixative was injected ventrally into the last abdominal segment over a 12 min period after removing the eyestalks and rostrum. After perfusion, the crayfish were dissected and tissues including gills, stomach, brain, and intestines were further immersed in precooled fixative and trimmed into small pieces.

### 2.4. Ultrastructural Observation and Electron Tomography

For morphological observations, the tissues were further subjected to a standard protocol of post-fixation (1% Osmium tetroxide in 0.1 M PB for 90 min), dehydration, and embedding (Spurr’s medium) after overnight fixation at 4 °C. Semi-thin sections (500 nm) were cut for toluidine blue O (TBO) stain and optical microscopic examinations. Ultra-thin sections were collected on 150-mesh copper grids and stained with 10% uranyl acetate (UA) in methanol (20 min) and Reynold’s lead citrate (LC, 4 min). The virus-infected nuclei were confirmed using a TEM (FEI Tecnai G2 TF20 Super TWIN) operating at 120 kV. Two-dimensional images were collected in the area without obvious artifacts. The blocks with acceptable structural preservations were further processed for electron tomography.

For electron tomography, the blocks were further trimmed and the serial sections (200 nm) through the target area were obtained as previously described [18]. Double-tilt electron tomography was performed with an FEI Tecnai TEM operating at 200 kV. The 3D volume of WSSV-infected nuclei was reconstructed, combined, and joined with eTomo [19]. The datasets with abundant sheets and assembling particles were chosen for manual segmentation. For the segmentation, the tomograms were passed through a Gaussian filter to remove the noise, and particles were manually selected based on the density thresholding in Amira. For the sheet-like structures, the aligned pieces in serial sections were joined. Viral particles were chosen only if their major parts were intact within a section.

### 2.5. Immuno-Gold Labeling

For immuno-gold labeling, small pieces of tissues were fixed in precooled fixative for 10 min and rinsed with cold 0.1 M PB three times (10 min each). The tissues were subjected to a standard protocol of dehydration without post-fixation. The dehydrated samples in ethanol were incubated with 0.1% UA in acetone for 8 h at −20 °C in a homemade freeze-substitution chamber. After rinsing with acetone three times (1 h each) at −20 °C, the specimens were subsequently infiltrated through an ascending gradient of Lowicryl HM20 resin (25%, 50%, 75%, 100%, and 100%, 2 h for each concentration) at −20 °C. The samples were further immersed overnight in 100% HM20. After the final change of 100% HM20 (1 h), the samples were embedded in a sample container the next day. The container was further sealed in a transparent air-tight box and the ultraviolet (360 nm) polymerization was set at −20 °C for 48 h. The container was then rewarmed to room temperature and ultraviolet radiation continued for another 24 h. Next, 100 nm sections of the tissues were prepared from the polymerized blocks and placed on 150-mesh nickel grids for immuno-gold labeling.

For labeling procedures, the sections on nickel grids were blocked with 5 % BSA in PBS for 20 min and incubated with the rabbit anti-VP664 (500×) [20], anti-VP28 (500×) [6], and anti-VP26 (500×) [6] antibodies in incubation buffer (1% BSA in PBS) for 2 h. Grids were subsequently washed with incubation buffer three times (10 min each). Secondary antibodies, goat anti-rabbit IgG (EM.GAR15, BB International, Cardiff, UK), with 15 nm gold conjugations at 20-fold dilution were further applied and incubated for 1 h. After washing with PBS, the sections were immersed in 1% glutaraldehyde in PBS for 5 min and washed three times with distilled water (10 min each). The negative controls of the staining were performed without the incubation of primary antibodies. UA and LC stains were performed as the counter stains. The specimens were inspected by TEM operating at 120 kV (FEI Tecnai G2 TF20 Super TWIN). 

## 3. Results

### 3.1. The Ultrastructure of the WSSV 

Since the ultrastructure analysis at the 2D level is still a common way to visualize the virus, here we confirmed the appearance of the virus using negative stain and ultra-thin sections (Figure 1). The 2D images of the WSSV virion showed a rod-like outer shape and the core was assembled by the stacked rings in the negative stain (Figure 1A,B). In infected cells, well-aligned virions could be found inside the expanded nuclei. Besides the aligned virions, some linear structures that were usually recognized as the nucleus capsid materials could also be observed (Figure 1D).

Since larger numbers of virus-infected nuclei were found in the intestines, serial sections and electron tomography on the intestines were performed to generate the serial tomograms (Videos S1–S4). In serial tomograms, the linear structures in 2D images were actually shown to be sheet-like structures (Figure 2). Those sheets usually came in pairs (Figure 2A,D). After segmentation, more associated structures could be found on the outer surface, while more regular patterns of molecules could be seen on the inner surface (Figure 2E,F). Occasionally, the sheets were accompanied by breaks or branches in 3D (Figure 2D–F). 

### 3.2. The Assembling Particles in WSSV-Infected Nucleus

The matured WSSV particles could be well arranged in the nucleus, while the area with immature particles could be found nearby. In our datasets, the immature particles could be classified into four different types (Figure 3). For the first type, the particles could be recognized by an empty core unit that was elongated and covered with a short piece of membranous structure (Figure 3A,B). The length of the elongated core could be greater than that of the matured particles. For the second type, several core units are sharing a longer piece of the membrane (Figure 3C,D). We observed that some of the core units were slightly elongated. For the third type, the electron-dense materials were found in the elongated core units (Figure 3E,F). Although some of the core units were not developed, the structures of elongated units were closer to those of the matured particles (Figure 3G,H).

### 3.3. The Protein Expression on the Virus-Associated Structures

To correlate the virus-associated structures with the protein assembling in matured virus particles, we checked the expressions of viral proteins with immuno-gold labeling (Figure 4). To confirm the specificity of antibodies for VP664, VP26, and VP28, positive signals on the matured particles were checked in our study. The gold particles for VP664 and VP26 were found inside the matured particles (Figure 4A,B), while the VP28 showed more abundant signals that surrounded the envelopes (Figure 4C). The gold particles were absent in the region without virus particles and no particles were observed in negative control sections (Figure 4D), suggesting the antibodies could be bound to the target protein specifically. 

The gold particles on virus-associated structures were then searched under TEM. Firstly, we checked protein expressions on the sheet-like structures (Figure 5). Although positive signals VP28 could be found near the sheets, the VP28 did not seem to be included in the main structural protein of the sheets. With VP26 absent on the sheets, only very rare bindings of the VP664 antibody were observed. Together with our staining results, our observations suggested that the sheets might not belong to a fully assembled structure in matured particles. 

Then, we focused our observations on the immature particles in the cell nucleus (Figure 6). Based on our 3D structural observations, the assembling particles could be recognized in 2D sections. In our observations, gold particles were found on the core units of the immature particles after VP664 or VP26 staining, suggesting that the core unit might be assembled by the tegument and nucleocapsid proteins. The positive signals of the VP28 were also observed on the membrane structures covering the cores, suggesting the membranous structures could be immature envelopes. Together with 3D structural reconstructions and immuno-gold labeling, the structure and composition of immature particles were identified in our study.

## 4. Discussion

As the key step in the viral life cycle, the assembling of virus particles could be a series of reactions among the protein subunits in the crowded intracellular spaces. During such reactions, the ordered and reproducible particles were generated to be infectious agents. Since the viruses could play a critical role in many animal diseases, understanding the assembling procedures could aid in the development of antiviral strategies [21]. In our study, the WSSV-associated structures in crayfish were reconstructed into 3D models. Additionally, the identity of the model components was further confirmed with immuno-gold labeling. The results of the current study could contribute to the knowledge of WSSV assembling steps in the cells.

Research reports that the capsids of viruses could be briefly classified into rod-like and spherical (polyhedron) structures [21]. As previously reported in the literature, the purified WSSV virions showed a rod-like nucleocapsid with envelopes after the routine negative stains in our observations [10,14,21,22]. As a large virus with long genomic DNA (about 300 kbp), the elongated and rod-like capsids of WSSV could provide enough spaces for the accommodations of its genetic materials [21,23]. For the nucleocapsid assembling in the rod-like viruses, the nucleic acids were usually surrounded by the capsids with helical symmetry [21]. However, the ring segments of WSSV capsids were recently demonstrated under an electron microscope after high-salinity and freeze-thaw treatments, suggesting different arrangements of capsid units in WSSV [14]. 

Since there are about 515–684 open reading frames that encode about 180 functional proteins (including more than 50 structural proteins) in the WSSV genome, the majority of viral protein functions may still call for investigations [5,23,24]. As the knowledge of WSSV assembly is quite limited due to its complexity, ultrastructural investigations on virus assembling have been performed in various cells and species [5,9,10,14,25,26]. However, the accumulation of structural knowledge through 2D structural observations could be limited. 

From early investigations in the WSSV-infected nucleus, a tubular structure of two electron-dense 20nm-wide longitudinal bands separated by a central 6nm-wide electron-clear band was reported in *Penaeus vannamei*, *P. stylirostris* [5,10,27], and *Penaeus monodon* [11]. The structure was considered to be the precursor of nucleocapsids [9,10] or viral nucleosomes [11]. Inconsistent with previous presumptions, our 3D structural investigations revealed a paired sheet-like structure in *Procambarus clarkii.* Although the existence of species- and tissue-specific structures after WSSV infection could not be denied, our observations suggested that the rod-shaped structures of the repeating subunits without the electron-lucent core mentioned in the literature could be an image of the paired sheets from the oblique projections [27]. On the other hand, the represented proteins for envelope, tegument, and nucleocapsids revealed no significant signals on the sheet-like structures in our study, suggesting the sheet-like structure might not belong to a matured structure of the virions. As the source of the sheets assembling was still unknown, although some associations between the sheets and VP28 (envelope) were noted, the role of the structure in WSSV infections remains to be investigated.

For the immuno-gold labeling of VP664, it is reported that the gold particles were not detected on the matured viruses with intact viral envelopes [20]. While the negative staining was performed to visualize the viral particles under TEM in that study, here the particles could be randomly orientated in resin-embedded tissues and the antigen could be exposed on the surface of the sections after ultrathin sectioning. So, the binding efficiency of the antibody and the patterns of the signals could be varied with sample preparation methods. 

Early 2D ultrastructural investigations have revealed the structures of immature WSSV virions in the center of the infected nucleus [8,9,10,11,12]. It was suggested that the envelopes could be bound to the immature WSSV virions during/after the assembling of the electron-dense materials in nucleocapsids [9,12,14]. Since the particle size of WSSV (120–150 nm in diameter; 270–290 nm in length) was larger than the ultrathin sections (70–90 nm in thickness), only parts of the immature particles could be available for conventional TEM observations. In our study, although the investigation areas were limited due to the throughput of sample processing and the manual segmentation for 3D visualizations, the 3D reconstruction from the semi-thin section (200 nm in thickness) would stand a better chance of obtaining the nucleocapsid with its associated structure for analysis. The envelope was evidenced on the least mature particles we obtained, suggesting the envelope-associated virion assembling could be significant in WSSV-infections. However, since the tissue- and species-specific structures could still exist, considering the limitations of the methods, more 3D structural studies are still necessary to clarify the process of WSSV assembling [13]. 

Recently, the knockdown of the envelope protein VP19 showed a connecting disruption between envelope and nucleocapsids of the WSSV particles in crayfish, where the nucleocapsids with matured size could still be obtained [15]. In our observations, although most of the assembling particles showed the nucleation and growth of the nucleocapsids on the envelopes, particles with grown nucleocapsids and smaller pieces of envelopes were also obtained. Since the low level of VP19 expression was still detectable, the WSSV nucleocapsids could be assembled independently from the envelope [9,14]. 

For the viral capsid formations, various mechanisms have been reported in the literature [21]. As the capsid assembling on nucleic acids or other polyelectrolytes was detected in some animal viruses [28,29,30], the capsid assembly for enveloped viruses could also occur during/before the envelope acquisitions [21]. While the binding between the viral envelopes and nucleocapsids could be established by electrostatic charges and/or hydrophobic moieties on protein-protein and protein-lipid interactions, the processes could be further assisted by cytoskeletal machinery or cellular factors to support membrane curvature [31,32,33,34]. In our study, the core units were first recognized by the surrounding membrane, and no intermediate structures (ex.: nicked core unit or free membrane) nor cellular components were noted in the nucleus with heavy virus accumulations. Currently, the information on the membrane source for WSSV assembling was still unclear. In our observations, the intranuclear networks of membranes were only recognized in one of the datasets (Appendix A). The membrane seems to connect with the nucleus envelope. However, currently, it is hard to judge the status of the host cell directly through structural observations within tissues, suggesting the 3D structural information at a specific timing could be important to researchers trying to uncover the procedures of viral replications [13].

## 5. Conclusions

The 3D structures of the WSSV-infected nucleus were reconstructed from *Procambarus clarkii* in this study. Our observation at the ultrastructural level revealed novel information about 3D structures after WSSV infections. The structures were also correlated to the matured virions through the expressions of viral proteins. However, more structural investigations on protein interactions and time-specific changes remain to be carried out to uncover the virus assembling procedures.

## Figures and Tables

**Figure 1 animals-12-01730-f001:**
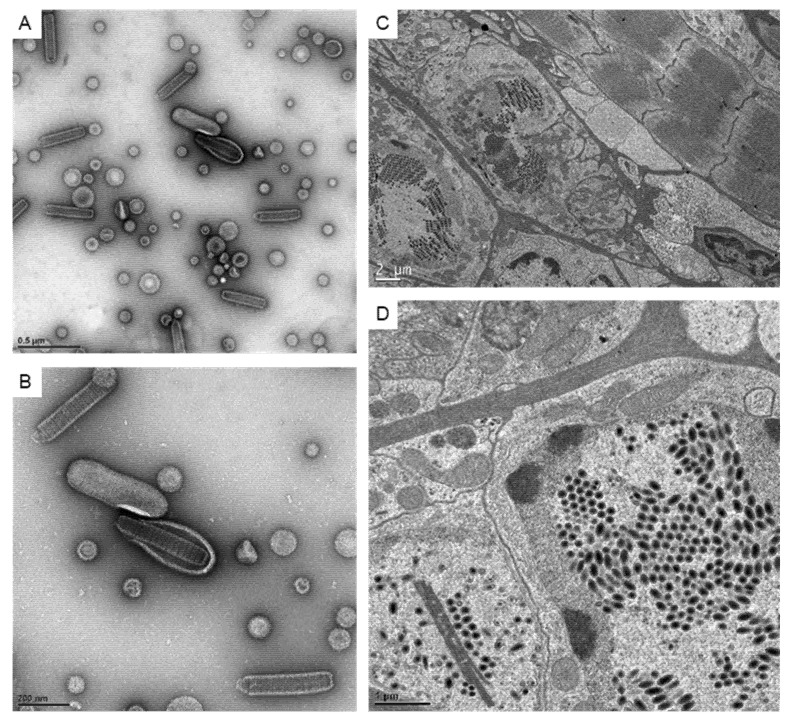
The images of WSSV particles and associated structures under TEM. (**A**,**B**): The represented images showing the negative stain of virus particles for crayfish inoculations. (**C**): The TEM images showing the WSSV-infected intestine tissues at low magnification. (**D**): The TEM image shows the virus particles and virus-associated structures in the nucleus.

**Figure 2 animals-12-01730-f002:**
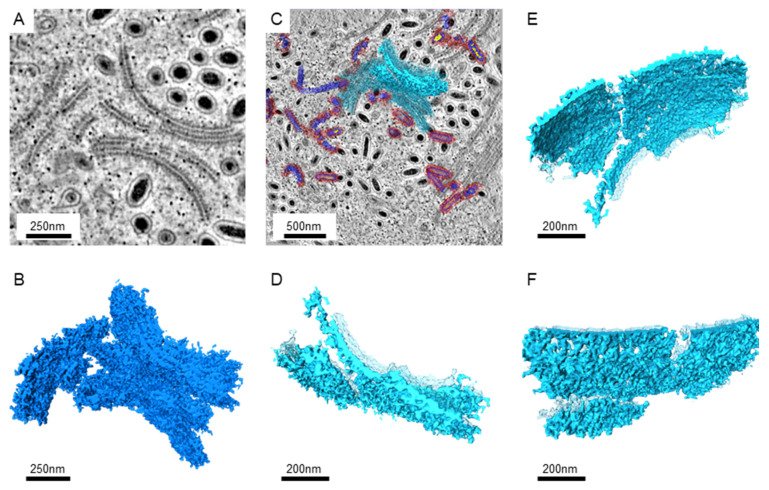
The images of the paired sheet-like structures in the WSSV-infected nucleus. (**A**): The representative image from the tomograms shows sheet pairs. The Gaussian filter was applied to reduce the noise. (**B**): The 3D structures of the sheet pairs were obtained from the serial sections near the area shown in A. The segmentation of the tomograms was performed manually with the threshold filter. (**C**): The image shows the position of the selected sheet structures near the area shown in A. The background image shows the cross-section at another z-axis level. The selected sheet is shown in solid azure color, while other sheets and segmented particles were shown in transparent colors. (**D**): The top view of the selected sheet is shown in high magnifications. The sheets without selection are shown in transparent azure color. (**E**): The side-view of the selected sheet shows the inner surface between the sheet pairs. (**F**): The side-view shows the outer surface of the selected sheet pair.

**Figure 3 animals-12-01730-f003:**
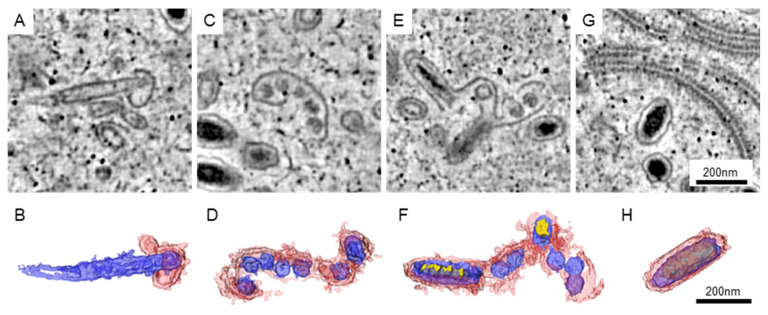
The immature particles in the WSSV-infected nucleus. The represented images of immature (**A**,**C**,**E**) and matured (**G**) particles in the tomograms are shown. (**B**,**D**,**F**,**H**) The 3D structure of the particles in (**A**,**C**,**E**,**G**, respectively). Red: the membrane-like structure presumed to be envelopes. Blue: the core units presumed to be the nucleocapsid and tegument. Yellow: the electron-dense materials presumed to be genetic materials.

**Figure 4 animals-12-01730-f004:**
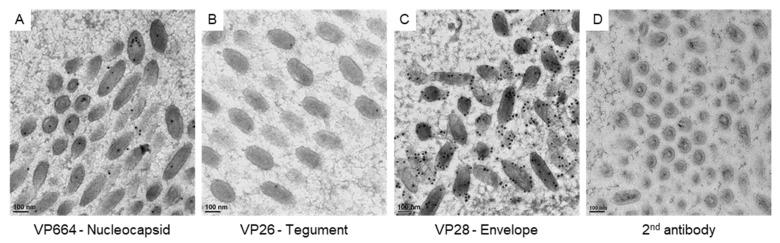
The expressions of viral proteins on matured WSSV particles. The represented images show the immuno-gold labeling of the nucleocapsid protein VP664 (**A**), the tegument protein VP26 (**B**), and the envelope protein VP28 (**C**) on matured WSSV particles. (**D**): the negative control without the incubation of primary antibodies.

**Figure 5 animals-12-01730-f005:**
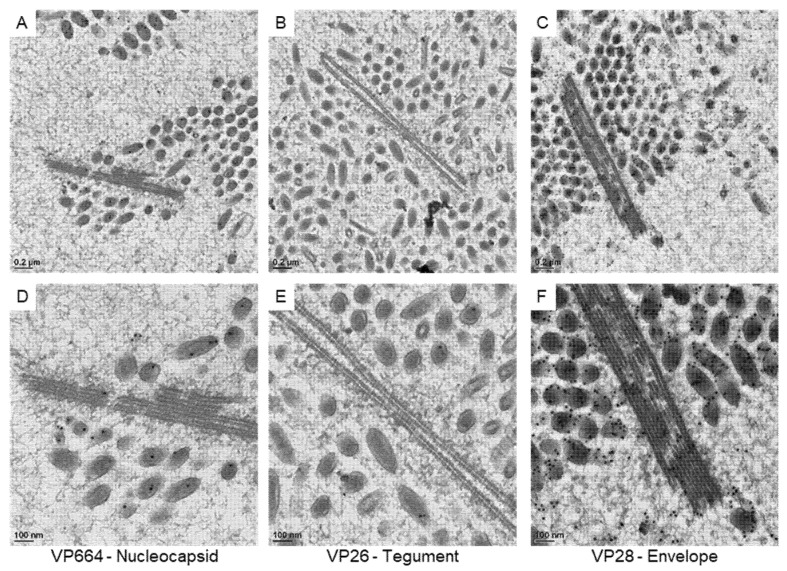
The expressions of viral proteins on the sheet-like structures in the WSSV-positive nucleus. The represented images show the immuno-gold labeling of the nucleocapsid protein VP664 (**A**), the envelope protein VP28 (**B**), and the tegument protein VP26 (**C**) in the nucleus. (**D**–**F**): The high magnification view of the sheet-like structure is shown in (**A**–**C**), respectively.

**Figure 6 animals-12-01730-f006:**
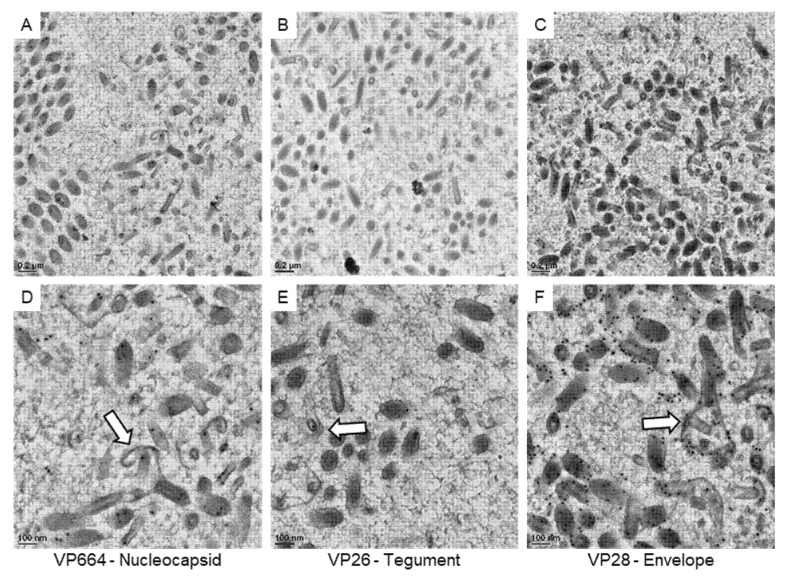
The expressions of viral proteins on the assembling WSSV particles. The represented images show the immuno-gold labeling of the nucleocapsid protein VP664 (**A**), the tegument protein VP26 (**B**), and the envelope protein VP28 (**C**) in the area containing assembling particles. (**D**–**F**): The high magnification view of assembling particles (the white arrows) found in the area of (**A**–**C**), respectively.

## Data Availability

The authors confirm that the data supporting the findings of this study are available within the article.

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
