# Peer review of "Three-Dimensional Investigations of Virus-Associated Structures in the Nuclei with White Spot Syndrome Virus (WSSV) Infection in Red Swamp Crayfish (Procambarus clarkii)"

_animals, 2022, doi:10.3390/ani12131730_

Round 1
Reviewer 2 Report
It is a very interesting and novel study about the 3D structures of the virus-associated structures and the viral particles types and their relationship with the localizations of viral proteins.
One question in material and methods section: 2.3.: How are you sure the crayfish selected for virus inoculation were really healthy?
In References section you might write the specific names in italic; por example citations: 11, 17 and 26.
Author Response
One question in material and methods section: 2.3.: How are you sure the crayfish selected for virus inoculation were really healthy?
Response: In our study, the captured crayfish were maintained for more than 2 generations in the lab. We didn’t monitor the pathogens in our culture system. However, the reproductive activities were normal. No abnormal behavior nor growth was observed in the population before inoculation. Usually, the decreased activities could be observed in the crayfish with WSSV infections, suggesting our crayfish were healthy before inoculation. We have included more information about the animal parts in material and methods (Line: 71-72).
In References section you might write the specific names in italic; por example citations: 11, 17 and 26.
Response 2: The correction has been made (Line: 375, Line: 383, Line: 396, and Line: 420).

Reviewer 3 Report
The written English of this manuscript needs considerable revision before I will review this for content.
Author Response
Thank you for your comment. The written English of the manuscript has been edited.

Round 2
Reviewer 1 Report
The revised version now generally in good structure.
Author Response
We deeply appreciate your review and comments. Those suggestions have been very helpful to us. Thank you.